# Use of Bacteria to Activate Ground-Granulated Blast-Furnace Slag (GGBFS) as Cementless Binder

**DOI:** 10.3390/ma15103620

**Published:** 2022-05-18

**Authors:** Woo Sung Yum, Jinung Do

**Affiliations:** 1Research Institute for Safety Performance, Korea Authority of Land and Infrastructure Safety (KALIS), Jinju 52856, Korea; wsyum@kalis.or.kr; 2Department of Ocean Civil Engineering, Gyeongsang National University, Tongyeong 53064, Korea

**Keywords:** GGBFS, bacteria, cementless binder, XRD, TG/DTG, MIP, water absorption rate

## Abstract

Ground-granulated blast-furnace slag (GGBFS) can be used as a cementless binder after activation. Recent approaches to activate GGBFS have focused on chemical methods that use NaOH, KOH, and CaO. This study introduces the use of bacteria to activate GGBFS as a biological approach. The presence of bacteria (volumetric ratio), curing temperature (23 °C and 60 °C), and number of curing days (3, 7, and 28 d) are investigated. The use of urea is considered owing to the possibility of calcium carbonate formation. The activated GGBFS is evaluated in the form of a cube (5 cm × 5 cm × 5 cm) for its strength, mineral identification, and pore size distribution. A brick (19 cm × 9 cm × 5.7 cm) is prefabricated to see the feasibility of commercializing bacteria-activated GGBFS based on water absorption and strength measurements. All results are compared with those of water-activated GGBFS. The results indicate that the use of urea inhibits the strength improvement of bacteria-activated GGBFS. Bacterial suspension enhances the GGBFS strength at a curing temperature of 60 °C. Mineral identification tests show that the strength increase is primarily due to the formation of calcite. The compressive strength satisfies the commercial standard of concrete bricks; however, the water absorption rate must be resolved.

## 1. Introduction

Global warming has consistently threatened human life through the possibility of abnormal environmental phenomena, such as extreme heat or cold waves, hurricanes, torrential rain, drought, and sea-level increases [1,2]. Through the Paris Agreement in 2015, global communities have agreed to maintain the global average temperature increase below 2 °C, as compared with pre-industrial levels, and limit the temperature increase to 1.5 °C or less. Based on the agreement, each nation has set a target degree for greenhouse gas reduction. Significant efforts are being expended in various industrial fields to reduce greenhouse gas emissions. Synthetic cement is a problematic material associated with global warming. One ton of CO_2_ is generated by manufacturing one ton of cement [3], and the cement industry is the main source of global warming [4,5,6].

Researchers have been developing cementless binders to replace synthetic cement with CO_2_-free cementing agents, such as fly ash, wood ash, lime, gypsum, red mud, eggshells, and rice husks [7,8,9,10,11,12,13]. Ground-granulated blast-furnace slag (GGBFS) is a byproduct of iron and steel production and is one of the most well-known cementless binders [14,15,16]. GGBFS is primarily composed of CaO and SiO_2_ (i.e., calcium silicate hydrates, C–S–H), which endow it with high strength. The cementation characteristics of GGBFS are realized using activators such as NaOH, Na_2_O, KOH, and CaO [17,18,19,20,21,22,23]. Activated GGBFS affords cementation without the use of synthetic cement; however, the chemical activators consume energy during the production process. Therefore, more eco-friendly activators must be identified to achieve a better sustainable use of GGBFS.

Bacteria constitute ~15% of the entire biomass on Earth, whereas human beings constitute ~0.01% [24]. Recently, the use of microbes in construction and building materials has garnered worldwide attention [25,26]. Bacteria can be used to enhance the strength and stiffness of concrete. Self-healing concrete is the most well-known concept for the utilization of microbes in concrete [27,28,29,30]. The fundamental principle of self-healing concrete is the use of ureolytic bacteria to hydrolyze urea-producing carbonate ions, as expressed in Equation (1).
(1)$$\mathrm{CO}{\left({\mathrm{NH}}_{2}\right)}_{2}+2H_{2}O\overset{\mathrm{bacteria}}{\to }2{\mathrm{NH}}_{4}^{+}+{\mathrm{CO}}_{3}^{2-}$$

The carbonate produced induces the formation of carbonate-based minerals that cause cementation [31]. GGBFS contains a sufficient number of divalent cations (e.g., Ca^2+^, Mg^2+^, etc.). If bacteria and urea are added to GGBFS, then the produced carbonate precipitates calcium carbonate or magnesium carbonate, etc. In fact, bacteria may activate the GGBFS because bacterial surfaces are negatively charged, which can alter the inherent characteristics of GGBFS. The activation of GGBFS using chemical activators has been investigated in several studies [17,18,19,20]; however, studies regarding the activation of GGBFS using natural activators, particularly microbial activators, are limited.

The aim of this study is to evaluate bacteria-activated GGBFS. The number of bacteria, curing temperature, and number of curing days are analyzed and compared with those of water-activated GGBFS. Urea is used to examine whether it induces the precipitation of calcium carbonate. The samples are cured in a 5 cm^3^ mold; subsequently, their strength, mineralogy, and pore size are analyzed. A bacteria-activated GGBFS brick is cured in 190 mm × 90 mm × 57 mm molds to determine the possibility of commercialization.

## 2. Materials and Methods

### 2.1. GGBFS

Commercial GGBFS was used in this study (Chunghae material Co., Ltd., Gwangyang, Korea). The grain size of the GGBFS was analyzed based on ASTM D6913-17 [32], and the grain size distribution is shown in Figure 1. All the grains measured between 0.5 and 60 µm. The mean grain size (50% cumulative distribution) is approximately 10 µm.

X-ray diffraction (XRD) and X-ray fluorescence (XRF, S8 Tiger wavelength dispersive WDXRF spectrometer, Bruker, Billerica, MA, USA) were conducted to evaluate the elemental constituents and mineralogy of the GGBFS. The XRD patterns of the GGBFS were obtained using a high-power powder X-ray diffractometer (D/Max2500V/PC, Rigaku, Japan) with Cu Kα radiation (k = 1.5418 Å). The measured XRD patterns were analyzed using the X’pert HighScore Plus software based on the International Centre for Diffraction Data PDF-2 database and the Inorganic Crystal Structure Database [33]. The XRF pattern shows that the GGBFS is primarily composed of oxides. Table 1 summarizes the oxide composition of the GGBFS; in particular, CaO, SiO_2_, and Al_2_O_3_ constituted 41.06%, 39.48%, and 12.35% by mass, respectively. The XRD results confirmed that the GGBFS is composed of akermanite, anhydrite, calcite, lime, glass, etc. (Figure 2).

### 2.2. Bacteria

An incubated bacterial suspension was used as the bacterial source in this study. Two milliliters of pre-incubated and frozen *Sporosarcina pasteurii* (ATCC 11859) was incubated in one liter of ammonium-yeast extract growth media (ATCC 1376) at 30 °C, and 200 rpm until a target bacterial density was obtained. The bacterial density was estimated by measuring the optical density at a wavelength of 600 nm (*OD*_600_) using a spectrophotometer (BKUV-1200, Konvision, South Korea). When the incubated bacterial suspension indicated *OD*_600_ ~1.0, the bacterial density was assumed to be ~10^7^ to 10^8^ cells/mL [34], and the incubation was terminated. The bacterial suspension was used immediately after incubation. For urea, a commercial product was used (U1250, Sigma Aldrich, Germany).

### 2.3. Sample Preparation

The samples used in this study are presented in Table 2. The factors that affect the results include the presence of bacteria, presence of urea, curing temperature, and curing days. The GGBFS and urea are referred to as powders, whereas tap water and the bacterial suspension are classified as solutions. Urea measuring 0% or 2.5% *w*/*w* of the entire powder mass was used. Here, 2.5% urea is equivalent to 1.04 M. The amount of solution was fixed at 40% (*w*/*w*) of the entire powder mass. Three ratios of water to the bacterial solution were prepared: 4:0, 3:1, and 2:2. Bacteria-dominant conditions (e.g., 1:3 or 0:4) were not considered, as the control of bacteria became excessive. The curing temperature was either 23 °C or 60 °C. Therefore, samples from 12 cases were selected. Herein, “U” denotes the presence of urea (2.5%), “B” indicates the amount of bacterial suspension (e.g., 10B = 10% bacterial suspension with 30% water), and 23 or 60 represents the curing temperature. For instance, U-10B at 60 refers 97.5% GGBFS and 2.5% urea, with 30% water and 10% bacterial suspension within the mass of GGBFS + urea, cured at 60 °C.

When preparing the sample, GGBFS and urea were mixed completely under dry conditions; subsequently, water and the bacterial solution were added to the powder and mixed. All the mixing procedures were performed based on ASTM C305-14 [35]. The mixed samples were poured into specific molds immediately after mixing was completed and then cured at the target temperature with 99% humidity. After completing the curing phase, the samples were treated before they were tested using a solvent substitution method and vacuum drying to prevent further reactions [36].

### 2.4. Compressive Strength

The mixed sample was poured into a 125 cm^3^ cubic mold to measure its compressive strength. The compressive strength of the sample was evaluated after 3, 7, and 28 days of curing at 23 °C or 60 °C (UH-F500kNX, Shimadzu, Kyoto, Japan, at a speed of 0.4 mm/min). The measurements were triplicated and averaged based on ASTM C109/C109M-02 [37].

### 2.5. Mineralogy Identification

After the compressive strength was determined, the samples were finely ground for XRD and thermogravimetric analysis to determine the mineralogy of the sample. In this regard, the XRD methodology described in Section 2.1 was used. Thermogravimetric characteristics can provide unique information regarding the mineralogy of a sample. Thermogravimetry (TG) was performed using a thermal analyzer (SDT Q600, TA Instruments, New Castle, DE, USA) equipped with alumina pans. The heating temperature was set from 30 to 1000 °C at a heating rate of 10 °C/min in a nitrogen gas environment. The gravitational mass change was recorded over time. Yum et al. [38,39,40] discovered that thermogravitational characteristics can be analyzed more accurately by differentiating the data from TG (e.g., mass over temperature), which is known as differential thermogravimetry (DTG). Therefore, both TG and DTG were employed in this study (i.e., TG/DTG).

### 2.6. Pore Size Distribution

A sample was prepared using a 5 mm^3^ mold to measure the pore size distribution of bacteria-activated GGBFS using mercury intrusion porosimetry (MIP) (Auto pore IV 9500, Micrometrics Instrument Co., Norcross, GA, USA). The samples were immersed in isopropanol prior to measurement. A pressure of 414 MPa (e.g., 60,000 psi) with a contact angle of 130° was applied to introduce mercury into the sample. The total porosity and mean pore diameter were evaluated.

### 2.7. Water Absorption Rate

The water absorption rate of the bacteria-activated GGBFS was measured to determine its potential as a bio-brick. The water absorption rate is calculated as the mass ratio between dry and wet conditions [41]. A mold measuring 190 mm × 90 mm × 57 mm was used to prepare samples for the experiment; subsequently, the samples were cured for 3 days at 60 °C (i.e., the CON 60 group). The curing day was based on the worst scenario during the sample preparation. After measuring the water absorption rate, the compressive strength of the samples was measured to investigate their strength while considering the water absorption effect.

## 3. Results

### 3.1. Compressive Strength

The compressive strengths of the samples are shown in Figure 3. The compressive strengths varied with the number of bacteria, presence of urea, and curing temperature. For the CON 23 group (no urea and cured at 23 °C, Figure 3a), the compressive strength of the samples decreased as the number of bacteria increased. Meanwhile, a sample of 20B at 23 was uncemented after 3 days of curing owing to its imperceptible compressive strength. This is attributed to the reduction in water required for the hydration reaction of the GGBFS. Mehta and Monteiro [3] proposed an optimal water-to-cement ratio (W/C) of 0.43; however, sample 20B at 23 exhibited an effective W/C of 0.2, except for the added bacterial suspension. Second, bacteria under ambient conditions may inhibit the hydration reaction of the GGBFS following a reduction in the compressive strength [42]. Detailed explanations are provided later along with other analyses.

Meanwhile, the CON 60 group (no urea and cured at 60 °C, Figure 3b) exhibited a compressive strength pattern different from that of the CON 23 group. The greater the amount of bacterial suspension added, the higher the compressive strength. In general, the compressive strengths of the CON 60 group were two to three times higher than those of the CON 23 group. Interestingly, the compressive strength of the CON 60 group samples did not increase significantly after 3 days of curing. For example, 0B at 60 exhibited a gradual increase in compressive strength, i.e., 17.79, 18.85, and 20.63 MPa after 3, 7, and 28 days of curing, respectively. By contrast, 10B at 60 and 20B at 60 showed almost negligible increases, i.e., 21.33, 21.41, and 21.23 MPa, and 23.22, 24, and 23.9 MPa, respectively. This observation can be interpreted as hydration at high temperature causing a rapid reaction following a rapid increase in the initial strength that subsequently stabilized. Additionally, comparing the CON 60 and CON 23 groups, the bacterial suspension at 23 °C appears to have inactivated the GGBFS, as compared with that at 60 °C.

When urea was added to the samples (Figure 3c,d), the general trends of the compressive strengths were similar to those without urea (Figure 3a,b), but the compressive strength was approximately 20% lower. Both the curing temperatures of 23 °C and 60 °C yielded similar trends for the samples with and without urea. Urea was assumed to be the byproduct of carbonate from urea hydrolyzed by bacteria. The hydrolysis of urea improves at moderate temperatures, high pH levels, and in the presence of high ureolytic bacteria [43,44]. The results show that the added urea inhibited the increase in the compressive strength of the bacteria-activated GGBFS system. If the effects of temperature and pH are excluded, then the mixed bacteria may not hydrolyze urea in the mixture. This is likely due to the high viscosity of the mixture and the small pore size of the mixture (this will be discussed in the analysis of the pore size distribution). The environment was unconducive to the movement of bacteria and hence their reaction with urea. Therefore, the use of urea is ineffective for improving the engineering properties of bacteria-activated GGBFS.

### 3.2. Identifications of Associated Minerals

#### 3.2.1. XRD

The XRD results are shown in Figure 4 and Figure 5. The XRD result for 20B at 23 for 3-day curing is omitted because of uncementation (Figure 4b). Mineralogy identification by XRD confirmed that calcium–silicate–hydrate (C–S–H), ettringite, and calcite were present in all the samples regardless of the presence of bacteria, presence of urea, and curing temperature (Figure 4 and Figure 5). The presence of C–S–H, ettringite, and calcite enhanced the cementation of the material. The results confirmed that no new product was produced even when urea was added. Considering the effect of urea on strength, as shown in Figure 3, the presence of urea impaired the strengthening of bacteria-activated GGBFS without transforming the reaction product.

#### 3.2.2. TG/DTG

The TG/DTG results are summarized in Figure 6 and Figure 7. The TG/DTG results for 20B at 23 and U-20B at 23 for 3-day curing are omitted because of uncementation (Figure 6a and Figure 7a). All the TG/DTG results reveal the presence of C–S–H, ettringite, and calcite, regardless of the curing temperature and the presence of bacteria and urea, which is consistent with the XRD analyses. The CON 23 group indicated higher amounts of C–S–H and ettringite and lower amounts of calcite in sample 0B than in samples 10B and 20B. The bacterial suspension resulted in higher calcite precipitation, which affected the strength [45,46]. The bacterial suspension contained bacteria, yeast, and ammonium sulfate meaning no calcium or carbonate ions. Therefore, the added bacterial suspension can activate the GGBFS to form calcite; however, the exact mechanism by which the bacterial suspension causes calcite formation requires further investigation.

The CON 60 group exhibited a pattern similar to that of the CON 23 group, i.e., C–S–H, ettringite, and calcite are indicated (Figure 6c,d). In pure GGBFS, C–S–H and ettringite are the primary minerals associated with its strength. When a greater amount of bacterial suspension was added, a higher amount of calcite was measured, and a higher strength was yielded (Figure 3b,d). Therefore, the activation of GGBFS through bacteria is primarily afforded by calcite formation.

In the case of the U-CON 23 for 3 days of curing, the amounts of C–S–H and calcite were similar regardless of whether bacterial suspension was added (Figure 7a). This indicates that the added urea inhibited the formation of calcite, resulting in no strength improvement. The results of the U-CON 23 group for 28 days of curing indicate low amounts of C–S–H, ettringite, and calcite (Figure 7b). Therefore, it can be concluded that urea inhibits the activation of GGBFS by the bacterial suspension.

The U-CON 60 sample after 3 and 28 days of curing showed greater formations of C–S–H and ettringite (Figure 7c,d). When urea was added at a curing temperature of 60 °C, C–S–H and ettringite contributed the most significantly to the strength. Meanwhile, calcite was the primary factor affecting strength improvement when no urea was added at a curing temperature of 60 °C. Therefore, the mechanism of strength improvement differs with the presence of urea in bacteria-activated GGBFS.

### 3.3. Pore Size Distribution

The pore size distribution of the samples provides valuable insights into bacteria-activated GGBFS. The MIP results are summarized in Figure 8 and Figure 9. A larger pore diameter results in a lower strength [47]. A higher amount of added bacterial suspension resulted in a larger pore diameter when the curing temperature was 23 °C and no urea was added (e.g., CON 23 group, Figure 8a,b). At a higher curing temperature (e.g., CON 60 group, Figure 8c,d), the pore diameter and total porosity decreased as the amount of bacterial suspension increased. The decrease in porosity resulted in a higher strength in the samples. Based on the TG/DTG observations, the precipitated calcite rendered the sample denser. Hence, a curing temperature of 60 °C induced active hydration between the bacterial suspension and GGBFS, resulted in greater calcite formation, and rendered the sample denser owing to its greater strength.

The U-CON 23 and U-CON 60 samples showed trends similar to those of the CON 23 and CON 60 samples (Figure 9). However, when urea was added, a relatively higher porosity was measured compared with when no urea was added. Urea dissolves in water, forming a strong hydrogen bond. The dissolved urea occupies a larger space in the samples without hydration, resulting in higher porosity. This may have contributed to the lower strength measurements.

### 3.4. Possibility of Commercializing Bacteria-Activated GGBFS

In addition to strength, water absorption characteristics are vital to the commercialization of construction materials. As a pre-fabricated brick, cubic specimens (190 mm × 90 mm × 57 mm) prepared based on the CON 60 group were cured for 3 days and assessed in terms of their water absorption rate and compressive strength. First, a visual difference was observed on the surface of the bricks depending on the recipe (Figure 10). The brick prepared using the 0B recipe appears bright gray (Figure 10a), whereas it transforms into navy as the bacterial suspension was added (Figure 10b,c). Aghaeipour and Madhkhan [48] reported that the complex reaction of sulfide with GGBFS was attributed to blue-green coloration. The presence of sulfate in the bacterial suspension resulted in a color change in the GGBFS brick.

Table 3 summarizes the water absorption rates and compressive strengths of the bricks. The greater the amount of bacterial suspension added, the lower the water absorption and the higher the compressive strength. Additionally, the quality specifications of the concrete bricks are provided in Table 3 [41]. Type I concrete brick is used for indoor/outdoor structures under pressure, whereas Type II concrete brick can be used for indoor structures under no pressure. Every GGBFS brick satisfied the compressive strength standard (e.g., 15.54–20.15 MPa, which is >8–13 MPa). However, only the water absorption rate of the 20B specimen (12.08%) satisfied the standard of Type II concrete brick (13%). Because all specimens satisfied the strength restriction, the bacteria-activated GGBFS brick can be used in a manner similar to Type II concrete brick if the water absorption characteristic is resolved.

## 4. Conclusions

In this study, the possibility of activating GGBFS using bacteria was investigated. Various factors, such as the ratio of bacteria, presence of urea, and curing temperature, were evaluated. The strength, mineral identification, pore size distribution, and water absorption rate were quantified based on different recipes. The findings of this study are as follows:A hypothesis was formed, wherein using bacteria and urea would hydrolyze urea and subsequently induce carbonate ions, which would consequently form calcite with calcium in GGBFS. However, the results indicated that the use of urea inhibited the strength improvement of bacteria-activated GGBFS. Hence, it was inferred that hydrated urea was not hydrolyzed because bacteria could not move freely to consume the urea within the GGBFS binder.Urea was hydrated but not ionized in the bacteria–GGBFS mixture. Hydrated urea occupied the pore space of the bacteria–GGBFS mixture, resulting in a higher porosity and lower strength compared with those of the sample without urea.The presence of bacterial suspension at a curing temperature of 23 °C was not conducive to the strength improvement of the GGBFS. However, incorporating bacterial suspension to the GGBFS at a curing temperature of 60 °C resulted in a higher strength as compared with one not incorporating bacterial suspension.The mineralogical identification of the bacteria-activated GGBFS indicated calcite formation as the primary contributor to the strength improvement.The strength of the bacteria-activated GGBFS was sufficient for the construction of bricks; however, the water absorption rate must be addressed for the successful commercialization of bacteria-activated GGBFS.

## Figures and Tables

**Figure 1 materials-15-03620-f001:**
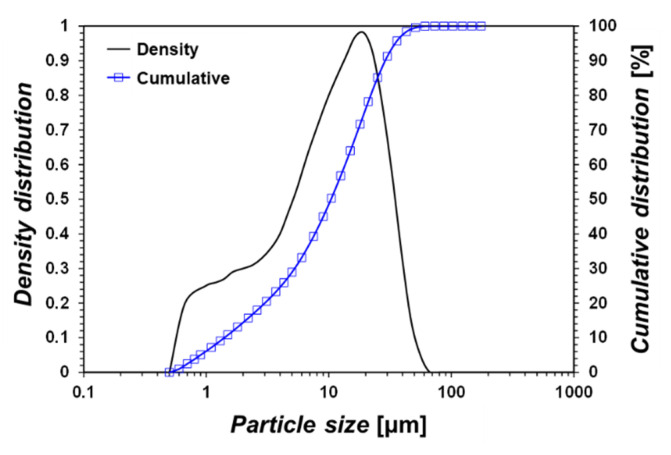
This is a figure. Schemes follow the same formatting.

**Figure 2 materials-15-03620-f002:**
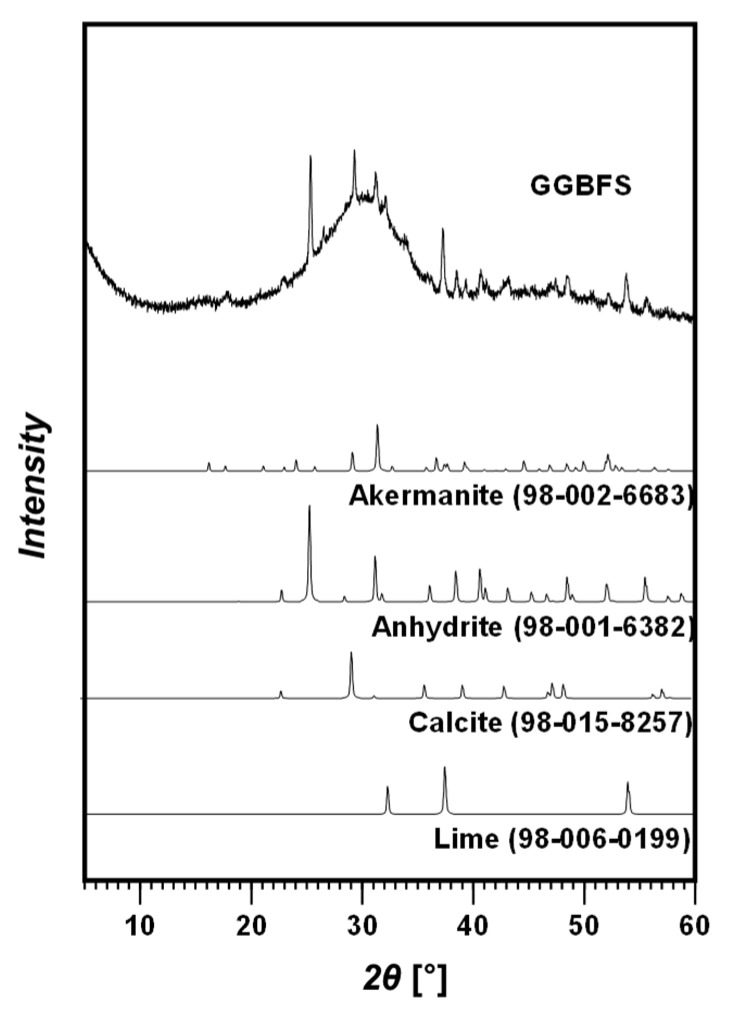
XRD patterns of GGBFS with identified phases.

**Figure 3 materials-15-03620-f003:**
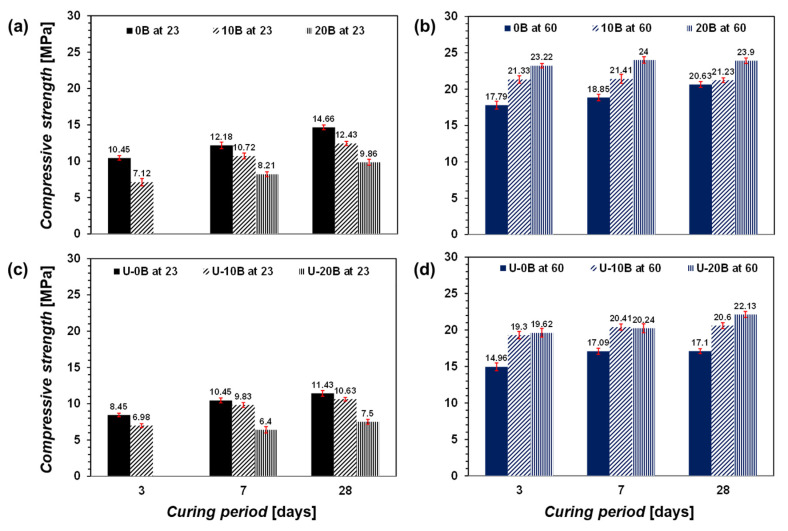
Compression test results: (**a**) CON 23 group, (**b**) CON 60 group, (**c**) U-CON 23 group, and (**d**) U-CON 60 group.

**Figure 4 materials-15-03620-f004:**
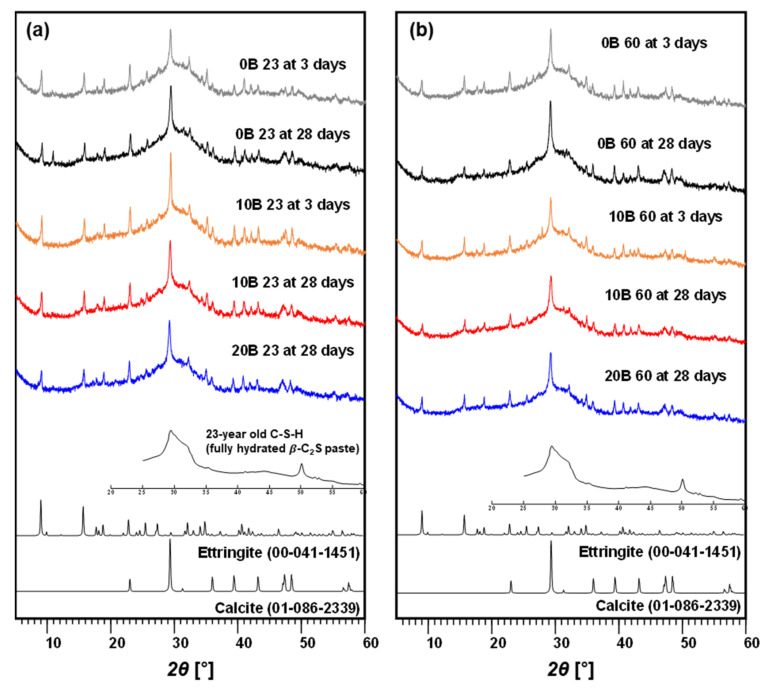
XRD results: (**a**) CON 23 group and (**b**) CON 60 group.

**Figure 5 materials-15-03620-f005:**
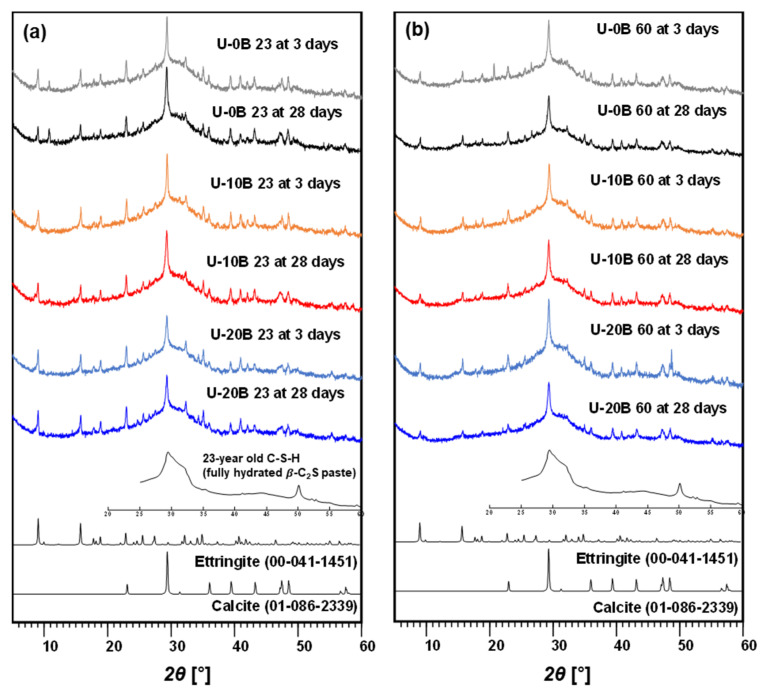
XRD results: (**a**) U-CON 23 group and (**b**) U-CON 60 group.

**Figure 6 materials-15-03620-f006:**
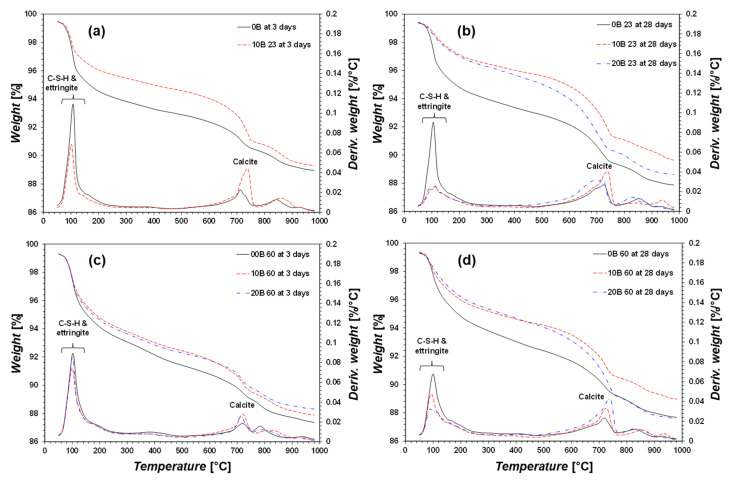
TG/DTG results: (**a**) CON 23 for 3-day curing, (**b**) CON 23 for 28-day curing, (**c**) CON 60 for 3-day curing, and (**d**) CON 60 for 28-day curing.

**Figure 7 materials-15-03620-f007:**
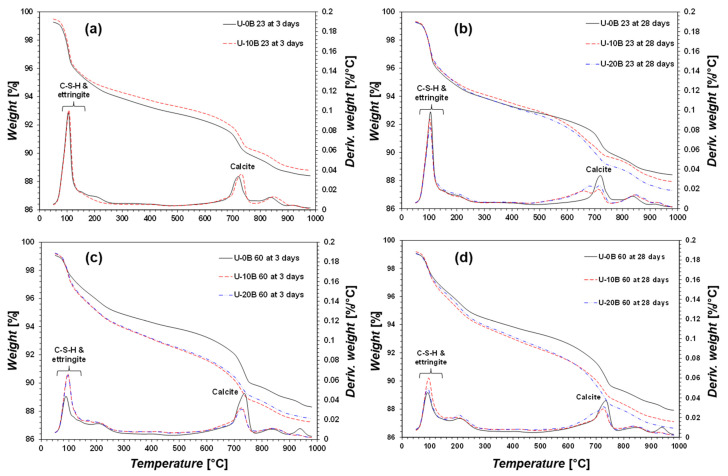
TG/DTG results: (**a**) U-CON 23 for 3-day curing, (**b**) U-CON 23 for 28-day curing, (**c**) U-CON 60 for 3-day curing, and (**d**) U-CON 60 for 28-day curing.

**Figure 8 materials-15-03620-f008:**
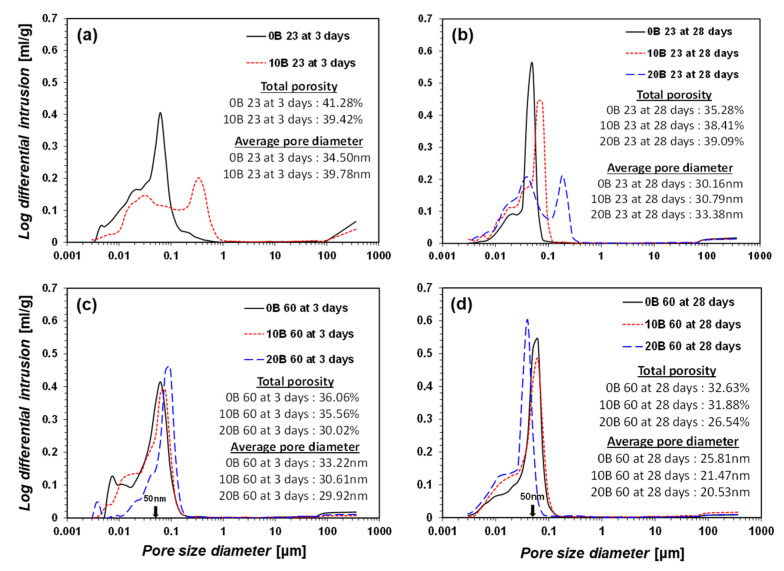
MIP results: (**a**) CON 23 for 3-day curing, (**b**) CON 23 for 28-day curing, (**c**) CON 60 for 3-day curing, and (**d**) CON 60 for 28-day curing.

**Figure 9 materials-15-03620-f009:**
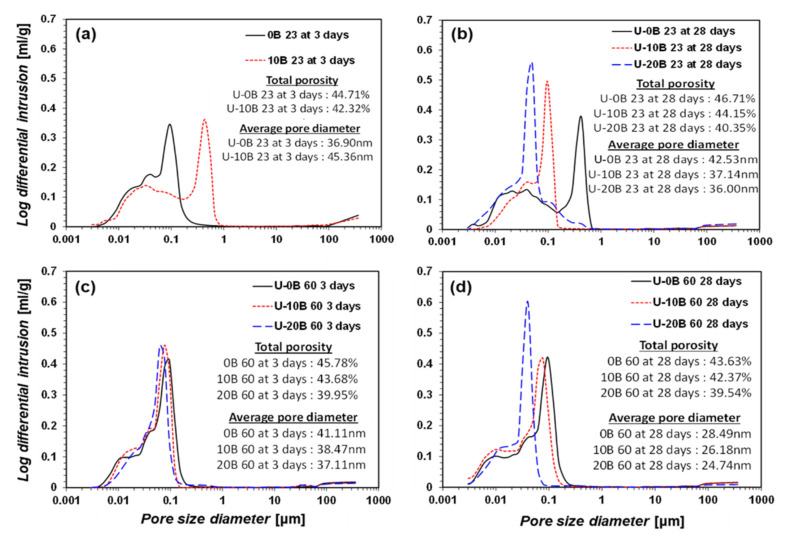
MIP results: (**a**) U-CON 23 for 3-day curing, (**b**) U-CON 23 for 28-day curing, (**c**) U-CON 60 for 3-day curing, and (**d**) U-CON 60 for 28-day curing.

**Figure 10 materials-15-03620-f010:**
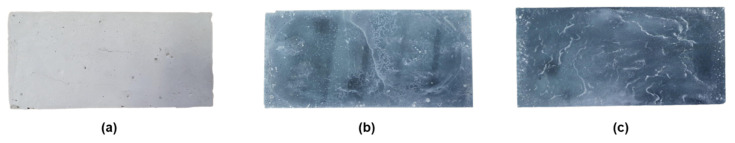
Plain view of bacteria-activated GGBFS bricks: (**a**) 0B, (**b**) 10B, and (**c**) 20B for 3-day curing at 60 °C.

**Table 1 materials-15-03620-t001:** Oxide compositions of GGBFS.

Oxide Compositions	CaO	SiO_2_	Al_2_O_3_	F_2_O_3_	SO_3_	MgO	K_2_O	TiO_2_	Na_2_O	MnO	Others
Percentage by mass (%)	41.06	39.48	12.35	0.39	2.9	2.16	0.43	0.77	0.15	0.15	0.16

**Table 2 materials-15-03620-t002:** Sample recipes and test conditions.

Name	Sample Group	Powders		Solutions		Curing Temperature (°C)
		GGBFS (% *w*/*w*)	Urea (%*w*/*w*)	Water (% per Powder)	Bacterial Suspension (% per Powder)	
0B at 23	CON 23	100	0	40	0	23
10B at 23				30	10	
20B at 23				20	20	
0B at 60	CON 60	100	0	40	0	60
10B at 60				30	10	
20B at 60				20	20	
U-0B at 23	U-CON 23	97.5	2.5	40	0	23
U-10B at 23				30	10	
U-20B at 23				20	20	
U-0B at 60	U-CON 60	97.5	2.5	40	0	60
U-10B at 60				30	10	
U-20B at 60				20	20	

**Table 3 materials-15-03620-t003:** Water absorption rate and compressive strength of bacteria-activated GGBFS bricks.

Case		Water Absorption Rate (%)	Compressive Strength (MPa)
Standard	Type I	Less than 7%	More than 13
	Type II	Less than 13	More than 8
Bacteria-activated GGBFS	0B	16.24	15.54
	10B	14.88	18.79
	20B	12.08	20.15

## Data Availability

The data presented in this study are available on request from the corresponding author.

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
