# Peer review of "Use of Bacteria to Activate Ground-Granulated Blast-Furnace Slag (GGBFS) as Cementless Binder"

_materials, 2022, doi:10.3390/ma15103620_

Round 1
Reviewer 1 Report
1. Line 61: Please provide references and review those studies to support the statement - "in several studies".
2. The innovation of this study should be emphasized. What are the knowledge gaps? How can this study bridge the gaps?
3. The literature review should be significantly improved. First, the alkali-activated slag can be reviewed with more details. The recent progress in using thermodynamic modelling to predict the hydration products of alkali-activated materials (including slag) should be included. "Analytical investigation of phase assemblages of alkali-activated materials in CaO-SiO2-Al2O3 systems: The management of reaction products and designing of precursors. Materials & Design, 194, p.108975."
Second, the more literatures and recent progress in bacteria-activated GGBFS should be reviewed.
4. The mechanism of bacteria-activated slag should be discussed considering the hydration products.
Author Response
Dear reviewer
The authors have revised the manuscript based on the reviewer's comments. Please check the attached. We hope the revised version provides improvement.

Reviewer 2 Report
The paper "Use of Bacteria to Activate Ground-Granulated Blast-Furnace Slag (GGBFS) as Cementless Binde" deals with a theme adherent to the theme of this journal and can be considered, after corrections:
(1) The abstract is short, relevant information is lacking, such as innovation, greater detailing of the methodology performed and the main results and general conclusion of the research, these points must be present;
(2) The introduction and literature review are short and should be improved, authors may consider running a table with other studies in the literature, in addition to adding some information about these residuals, such as: 10.1016/j.cscm.2021. e00845; 10.1016/j.cscm.2018.02.009; 10.1016/j.cscm.2021.e00723.
(3) The materials and methods section should be dedicated to showing the methodological steps, the authors mix it up by adding several results that should be in another section, in addition there is a need for greater detailing of various methodological procedures;
(4) The results and discussions section needs some comparisons with other previous studies, in addition several (or almost all) of the results presented have low quality of visualization, with bad texts, the authors should check this.
(5) Error bars must be checked on all results presented.
Author Response
Dear reviewer
We have revised the manuscript based on the reviewer's comments. Please check the attachement. We hope the revised version provides improvement.

Round 2
Reviewer 1 Report
The paper has been revised based on the comments.
Reviewer 2 Report
The authors have made all indicated corrections and the paper can be accepted at its current stage.